# Copula Multi-label Learning

**Weiwei Liu**
School of Computer Science, Wuhan University
Wuhan, China 430072
liuweiwei863@gmail.com

## Abstract

A formidable challenge in multi-label learning is to model the interdependencies between labels and features. Unfortunately, the statistical properties of existing multi-label dependency modelings are still not well understood. Copulas are a powerful tool for modeling dependence of multivariate data, and achieve great success in a wide range of applications, such as finance, econometrics and systems neuroscience. This inspires us to develop a novel copula multi-label learning paradigm for modeling label and feature dependencies. The copula based paradigm enables to reveal new statistical insights in multi-label learning. In particular, the paper first leverages the kernel trick to construct continuous distribution in the output space, and then estimates our proposed model semiparametrically where the copula is modeled parametrically, while the marginal distributions are modeled nonparametrically. Theoretically, we show that our estimator is an unbiased and consistent estimator and follows asymptotically a normal distribution. Moreover, we bound the mean squared error of estimator. The experimental results from various domains validate the superiority of our proposed approach.

## 1 Introduction

Multi-label learning [1, 2, 3, 4, 5, 6], which allows multiple labels for each instance simultaneously, is of paramount importance in a variety of fields ranging from protein function classification and video annotation, to automatic image categorization. For example, an image may have Cloud, Tree and Sky tags; labels such as Government, Policy and Election may be needed to describe the subject of the video; a gene can belong to the functions of Protein Synthesis, Metabolism and Transcription.

Binary relevance (BR) [7] is one of the most popular baselines for multi-label learning, which aims to independently train a binary classifier for each label. Recently, much of the multi-label learning literature [8, 9, 10, 11, 12, 13] have shown that the independent assumption among labels and features leads to degenerated performance. A plethora of methods have been motivated by a perceived need to modelling the dependence. For example, the classifier chain (CC) model [14] captures label dependency by using binary label predictions as extra input attributes for the following classifiers in a chain. CCA [15] uses canonical correlation analysis for learning label dependency. CPLST [10] uses principal component analysis to capture both the label and the feature dependencies. Unfortunately, the statistical properties of all these methods are still not well understood. This paper aims to fill this gap. Particularly, the work in this paper is inspired by Sklar's observation below.

**Sklar's Observation.** Sklar's Theorem [16] shows that the univariate margins and the multivariate dependence structure can be separated, and the dependence structure can be represented by a copula [17]. Therefore, a copula contains all the information that we need to measure dependence, and it is invariant to any nonlinear strictly increasing transformations of the marginal variables.

**Contributions.** Motivated by copulas's superiority in modeling dependence, we develop a novel copula multi-label learning paradigm for modeling label and feature dependencies. Our main

contributions may be summarized as follows. Firstly, we provide the statistical understandings for multi-label learning. Secondly, this paper leverages the kernel trick to construct continuous distribution in the output space, and then we use a semiparametric approach to estimate our proposed model where the copula is modeled parametrically, while the marginal distributions are modeled nonparametrically. Theoretically, we show that our estimator is an unbiased and consistent estimator and follows asymptotically a normal distribution. Moreover, we bound the mean squared error (MSE) of estimator. Our results show that the MSE of estimator goes to 0 as the number of training samples goes to infinity. The experimental results on several real-world data sets with a wide range of domains demonstrate that our proposed approach outperforms existing dependency modelings.

We organize this paper as follows. We first discuss the related work in the following paragraph. §2 presents some basic definitions and $n$-dimensional copula. §3 introduces copula multi-label learning paradigm and estimators. §4 presents the statistical properties of our proposed estimator, and experimental results are presented in §5. The last section provides our conclusions.

**Related Work.** Copula [17] modeling has become exceedingly popular in recent years, and has been successfully used in a diverse range of applications, especially in finance [18], actuarial science [19], survival analysis [20], systems neuroscience [21] and econometrics [22]. Due to its flexibility and simplicity in modeling the dependence, copulas have also been widely explored in machine learning community, such as domain adaptation [23, 24], Markov Chain Monte Carlo [25], latent markov networks [26], Bayesian networks [27], kernel learning [28] and graphical models [29, 30]. However, it is unclear whether copula is advantageous for multi-label learning. This work provides an affirmative answer.

## 2 Preliminaries

We denote by $DomH$ and $RanH$ the domain and range of a function $H$. Given two real vectors $\mathbf{a} = [a_1, \cdots, a_n]$ and $\mathbf{b} = [b_1, \cdots, b_n]$. We write $\mathbf{a} \leq \mathbf{b}$, if $a_i \leq b_i$ for all $i \in \{1, 2, \ldots, n\}$. For $\mathbf{a} \leq \mathbf{b}$, let $[\mathbf{a}, \mathbf{b}]$ represent the $n$-box $B = [a_1, b_1] \times \cdots \times [a_n, b_n]$, the Cartesian product of $n$ closed intervals. The vertices of an $n$-box $B$ are the points $\mathbf{d} = [d_1, \cdots, d_n]$, where $d_i$ is equal to either $a_i$ or $b_i$. We first present the following concepts and notations.

**Definition 1.** *A sequence of random variables $x_n$ is said to converge to a constant $\varpi$ in probability, in symbols $x_n \xrightarrow{P} \varpi$, if for every $\epsilon > 0$, $P(|x_n - \varpi| < \epsilon) \to 1$ as $n \to \infty$, or $P(|x_n - \varpi| \geq \epsilon) \to 0$ as $n \to \infty$.*

**Definition 2.** *Given two sequences of random variables $x_n$ and $y_n$. $x_n$ is of smaller order in probability than $y_n$, in symbols $x_n = o_p(y_n)$, if $\frac{x_n}{y_n} \xrightarrow{P} 0$. Particularly, $x_n = o_p(1)$, if and only if $x_n \xrightarrow{P} 0$.*

**Definition 3.** *Given two sequences of random variables $x_n$ and $y_n$. $x_n$ is of order less than or equal to that of $y_n$ in probability, in symbols $x_n = O_p(y_n)$, if given $\epsilon > 0$ there exists a constant $M = M(\epsilon)$ and an integer $m = m(\epsilon)$ such that $P(|x_n| \leq M|y_n|) \geq 1 - \epsilon$ for all $n > m$.*

**Definition 4.** *A sequence of random variables $x_n$ with cumulative distribution functions (CDF) $F_n$ converges in distribution to a random variable $x$ with CDF $F$, in symbols $x_n \xrightarrow{D} x$, if $\lim_{n \to \infty} F_n(x) = F(x)$, for all continuity points $x$ of $F$.*

Next, we introduce some important definitions before going to the copulas.

**Definition 5** ($H$-volume)**.** *Let $S_1, S_2, \cdots, S_n$ be nonempty subsets of $[-\infty, +\infty]$. Let $H$ be a real function of $n$ variables such that $DomH = S_1 \times \cdots \times S_n$. Let $B = [\mathbf{a}, \mathbf{b}]$ be an $n$-box all of whose vertices are in $DomH$. Then the $H$-volume of $B$ is given by $V_H(B) = \sum sgn(\mathbf{d})H(\mathbf{d})$, where the sum is taken over all vertices $\mathbf{d}$ of $B$, and $sgn(\mathbf{d})$ is given by*

$$sgn(\mathbf{d}) = \begin{cases} 1 & \text{if } d_i = a_i \text{ for an even number of } i\text{'s} \\ -1 & \text{if } d_i = a_i \text{ for an odd number of } i\text{'s} \end{cases}$$

**Definition 6** ($n$-increasing)**.** *A real function $H$ of $n$ variables is $n$-increasing if $V_H(B) \geq 0$ for all $n$-boxes $B$ whose vertices lie in $DomH$.*

**Definition 7.** *Suppose that the domain of a real function $H$ of $n$ variables is given by $DomH = S_1 \times \cdots \times S_n$, where each $S_i$ has a least element $a_i$. We say that $H$ is grounded if $H(\mathbf{d}) = 0$ for all $\mathbf{d}$ in $DomH$ such that $d_i = a_i$ for at least one $i$.*

If each $S_i$ is nonempty and has a greatest element $b_i$, then $H$ has margins, and the one dimensional margins of $H$ are the functions $H_i$ given by $DomH_i = S_i$ and $H_i(x) = H(b_1, \cdots, b_{i-1}, x, b_{i+1}, \cdots, b_n)$ for all $x$ in $S_i$. Higher-dimensional margins are defined by fixing fewer places in $H$. One dimensional margins are just called margins, and for $i \geq 2$, we will write $i$-margin for $i$-dimensional margin.

**Definition 8** ($n$-dimensional copula)**.** *An $n$-dimensional copula (or $n$-copula) is a function $C$ : $[0,1]^n \rightarrow [0,1]$ such that*

*(i) $C$ is $n$-increasing and grounded.*

*(ii) $C$ has margins $C_i$, $i \in \{1, 2, \ldots, n\}$, which satisfy $C_i(u) = u$ for all $u$ in $[0,1]$.*

Note that for any $n$-copula $C$, $n \geq 3$, each $i$-margin of $C$ is a $i$-copula, for $2 \leq i < n$. Next, we introduce the most important Sklar's Theorem regarding copulas.

**Theorem 1** (Sklar's Theorem [16])**.** *Let $H$ be an $n$-dimensional distribution function with marginal CDF $F_1, \ldots, F_n$. Then there exists an $n$-copula $C$ such that for all $(x_1, x_2, \ldots, x_n) \in [-\infty, +\infty]^n$,*

$$H(x_1, \ldots, x_n) = C(F_1(x_1), \ldots, F_n(x_n))$$

*If $F_1, \ldots, F_n$ are all continuous, then $C$ is unique; otherwise $C$ is uniquely determined on $RanF_1 \times \cdots \times RanF_n$. Conversely, if $C$ is an $n$-copula and $F_1, \ldots, F_n$ are distribution functions, then the function $H$ defined above is an $n$-dimensional distribution function with marginal CDF $F_1, \ldots, F_n$.*

Sklar's Theorem indicates that copula allows a complete separation of dependence modeling from the marginal distributions and by specifying a copula one can summarize all the dependencies between margins.

Assuming $H$ has $n$-order partial derivatives. Using the chain rule, we derive the joint density $f$ from the copula construction:

$$f(x_1, \ldots, x_n) = \frac{\partial^n C(F_1(x_1), \ldots, F_n(x_n))}{\partial F_1(x_1), \ldots, \partial F_n(x_n)} \prod_{i=1}^n f_i(x_i) = c(F_1(x_1), \ldots, F_n(x_n)) \prod_{i=1}^n f_i(x_i) \quad (1)$$

where $c(F_1(x_1), \ldots, F_n(x_n))$ is called the copula density function and $f_i$ is the marginal density function of $x_i$.

## 3 Copula Multi-label Learning

We denote the transpose of the vector/matrix by the superscript $'$ and the logarithms to base 2 by $log$. Let $|| \cdot ||_2$ represent the $l_2$ norm. Assume that $x = (x_1, \ldots, x_p)' \in \mathbb{R}^{p \times 1}$ is a random vector representing an input (feature), and $z = (z_1, \ldots, z_q)' \in \{0,1\}^{q \times 1}$ is a binary random vector representing the corresponding output (label) of multi-label learning (MLC). We denote by $F_a$ and $f_a$ the CDF and density function of $x_a$, $a \in \{1, \ldots, p\}$, respectively. Let $p_j$ represent the probability mass function of $z_j$, $j \in \{1, \ldots, q\}$.

### 3.1 Kernel Trick: Constructing Continuous Distribution

Note that the output of MLC is boolean-valued variable, and it is non-trivial to apply Sklar's Theorem to discrete variable. In this paper, we use a continuous distribution to replace each discrete distribution. In particular, we leverage a kernel density with a uniform kernel and a small bandwidth to construct the continuous distribution. Let $b$ be the bandwidth. $b$ should be less than or equal to half the distance of the two discrete points. For an observation at $z_j$, $j \in \{1, \ldots, q\}$, the kernel density function is $\frac{p_j(z_j)}{2b}$. By setting $b = 0.5$, we transform binary variable $z_j$ to continuous variable $y_j$ with CDF $\mathfrak{F}_j$:

$$\mathfrak{F}_j(y_j) = \begin{cases} 0 & y_j < -0.5 \\ p_j(0)(y_j + 0.5) & -0.5 \leq y_j \leq 0.5 \\ p_j(0) + p_j(1)(y_j - 0.5) & 0.5 < y_j \leq 1.5 \\ 1 & y_j > 1.5 \end{cases}$$

## 3.2 Copula Modeling

Using Sklar's Theorem, the CDF of $(y_1, \ldots, y_q, x_1, \ldots, x_p)$ can be expressed as $C(\mathfrak{F}_1(y_1), \ldots, \mathfrak{F}_q(y_q), F_1(x_1), \ldots, F_p(x_p))$, where $C$ is the $(p+q)$-copula function of $(y_1, \ldots, y_q, x_1, \ldots, x_p)$. In the following paper, we focus on the $(p+1)$-margin of $C$ on variables $(y_j, x_1, \ldots, x_p), \forall j \in \{1, \ldots, q\}$, which have inherited the label dependence information from $(p+q)$-copula function.

Using Eq.(1), we derive the conditional density of $y_j$ given $x$ as:

$$f_{y_j}(y_j|x) = \frac{\Upsilon_j(y_j)c(\mathfrak{F}_j(y_j), F_1(x_1), \ldots, F_p(x_p))}{c_x(F_1(x_1), \ldots, F_p(x_p))}$$

where $\Upsilon_j(y_j)$ is the density function of $y_j$ and $c_x(F_1(x_1), \ldots, F_p(x_p)) = \frac{\partial^p C(1, F_1(x_1), \ldots, F_p(x_p))}{\partial F_1(x_1), \ldots, \partial F_p(x_p)}$ is the copula density of $x_1, \ldots, x_p$. The conditional mean, $\Xi_j(x)$, of $y_j$ given $x$, can be derived as:

$$
\begin{aligned}
\Xi_j(x) &= E(y_j|x) \\
&= \int_{-\infty}^{+\infty} y_j f_{y_j}(y_j|x) dy_j \\
&= \int_{-\infty}^{+\infty} \frac{y_j \Upsilon_j(y_j) c(\mathfrak{F}_j(y_j), F_1(x_1), \ldots, F_p(x_p))}{c_x(F_1(x_1), \ldots, F_p(x_p))} dy_j \\
&= \frac{\vartheta_j(F_1(x_1), \ldots, F_p(x_p))}{c_x(F_1(x_1), \ldots, F_p(x_p))} \\
&= E\big(y_j \omega(\mathfrak{F}_j(y_j), F_1(x_1), \ldots, F_p(x_p))\big)
\end{aligned}
\tag{2}
$$

where $\omega(\mathfrak{F}_j(y_j), F_1(x_1), \ldots, F_p(x_p)) = \frac{c(\mathfrak{F}_j(y_j), F_1(x_1), \ldots, F_p(x_p))}{c_x(F_1(x_1), \ldots, F_p(x_p))}$ and $\vartheta_j(F_1(x_1), \ldots, F_p(x_p)) = E(y_j c(\mathfrak{F}_j(y_j), F_1(x_1), \ldots, F_p(x_p)))$.

Eq.(2) demonstrates that the conditional mean of $y_j$ given $x$ can be obtained from the copula density. It also indicates that the conditional mean is a weighted mean function, where the weights are induced by the copula function $\omega$ defined above.

**Proposition 1.** *If $p = 1$ or $x_1, \ldots, x_p$ are mutually independent, Eq.(2) reduces to $\Xi_j(x) = \vartheta_j(F_1(x_1), \ldots, F_p(x_p)), \forall j \in \{1, \ldots, q\}$.*

Given estimators $\widehat{\omega}$, $\widehat{\mathfrak{F}_j}$ and $\widehat{F_1}, \ldots, \widehat{F_p}$ for $\omega$, $\mathfrak{F}_j$ and $F_1, \ldots, F_p$, respectively, then $\Xi_j$ can be estimated by $\widehat{\Xi}_j(x) = E\big(y_j \widehat{\omega}(\widehat{\mathfrak{F}_j}(y_j), \widehat{F_1}(x_1), \ldots, \widehat{F_p}(x_p))\big)$. To estimate $\omega$, one needs estimators for the copula densities $c$ and $c_x$. Finally, we conduct the multi-label predictions based on the value of $\widehat{\Xi}_j(x)$. This paper uses a semiparametric approach where the copula is modeled parametrically, while the marginal distributions are modeled nonparametrically.

## 3.3 Estimators

Let $(y_1^{(i)}, \ldots, y_q^{(i)}, x_1^{(i)}, \ldots, x_p^{(i)}), i \in \{1, \ldots, n\}$, be $n$ independent and identically distributed (i.i.d.) training samples generated from the distribution of $(y_1, \ldots, y_q, x_1, \ldots, x_p)$. For $j \in \{1, \ldots, q\}$, the probability mass functions are estimated by $\widehat{p}_j(0) = \frac{\sum_{i=1}^{n} I(y_j^{(i)}=0)}{n}$ and $\widehat{p}_j(1) = 1 - \widehat{p}_j(0)$, where $I(\cdot)$ is the indicator function. $\mathfrak{F}_j$ is estimated by

$$
\widehat{\mathfrak{F}}_j(y_j) = \begin{cases}
0 & y_j < -0.5 \\
\widehat{p}_j(0)(y_j + 0.5) & -0.5 \leq y_j \leq 0.5 \\
\widehat{p}_j(0) + \widehat{p}_j(1)(y_j - 0.5) & 0.5 < y_j \leq 1.5 \\
1 & y_j > 1.5
\end{cases}
$$

We use kernel smoothing method to estimate $F_1, \ldots, F_p$. Let $k(\cdot)$ be a symmetric probability density function and $h$ be the bandwidth. For $j \in \{1, \ldots, p\}$, $F_j$ is estimated by

$$\widehat{F}_j(x_j) = \frac{\sum_{i=1}^{n} K\left(\frac{x_j - x_j^{(i)}}{h}\right)}{n}$$

where $K(t) = \int_{-\infty}^{t} k(u)du$. We make the following assumption for $\widehat{F}_j(x_j)$.

**Assumption A.** *For $j \in \{1, \ldots, p\}$,*

$$\widehat{F}_j(x_j) = \frac{\sum_{i=1}^{n} I(x_j^{(i)} \le x_j)}{n} + o_p(n^{-1/2})$$

We use a parametric approach to estimate the copula. Suppose that $(p + q)$-copula density belongs to a given parametric family $\mathbb{C} = \{c(\cdot; \theta), \theta \in \mathbb{R}^{v \times 1}\}$. Assume that $\theta^*$ is the true but unknown copula parameter. Maximum pseudo-likelihood estimator [31, 32] $\widehat{\theta}$ is one of the most popular estimators of $\theta^*$, which is defined as $\widehat{\theta} = \arg\max_\theta \log \sum_{i=1}^{n} c(\widehat{\mathfrak{F}}_1(y_1^{(i)}), \ldots, \widehat{\mathfrak{F}}_q(y_q^{(i)}), \widehat{F}_1(x_1^{(i)}), \ldots, \widehat{F}_p(x_p^{(i)}); \theta)$. Let $\theta_j^* \in \mathbb{R}^{d \times 1}$ and $\widehat{\theta}_j \in \mathbb{R}^{d \times 1}$ be the corresponding true and estimator of parameter for $(p + 1)$-margin on variables $(y_j, x_1, \ldots, x_p)$. We make the following assumption on $\widehat{\theta}_j$.

**Assumption B.** *For $j \in \{1, \ldots, q\}$,*

$$\widehat{\theta}_j - \theta_j^* = \frac{\sum_{i=1}^{n} \psi_i}{n} + o_p(n^{-1/2})$$

*where $\psi_i = \psi(\mathfrak{F}_j(y_j^{(i)}), F_1(x_1^{(i)}), \ldots, F_p(x_p^{(i)}); \theta_j^*)$ is a d-dimensional random vector such that $E(\psi) = (0, \ldots, 0)'$ and $E\|\psi\|_2^2 < \infty$.*

[32] shows that maximum pseudo-likelihood estimator satisfy this assumption.

The following analysis focuses on the $(p + 1)$-margin $c$ on variables $(y_j, x_1, \ldots, x_p)$. To simplify the analysis, we provide some simple notations. For $a \in \{1, \ldots, p + 1\}$, $c_a = \frac{\partial c}{\partial u_a}$. For $a \in \{1, \ldots, p\}$, $c_{x,a} = \frac{\partial c_x}{\partial u_a}$ and $\vartheta_{j,a} = \frac{\partial \vartheta_j}{\partial u_a}$. $\partial c_x = (c_{x,1}, \ldots, c_{x,p})'$. $\partial \vartheta_j = (\vartheta_{j,1}, \ldots, \vartheta_{j,p})'$. Let $\dot{c} = (\frac{\partial c}{\partial \theta_1}, \ldots, \frac{\partial c}{\partial \theta_d})'$, $\dot{c}_x = (\frac{\partial c_x}{\partial \theta_1}, \ldots, \frac{\partial c_x}{\partial \theta_d})'$ and $\dot{\vartheta}_j = (\frac{\partial \vartheta_j}{\partial \theta_1}, \ldots, \frac{\partial \vartheta_j}{\partial \theta_d})'$, where $\vartheta_j(u_1, \ldots, u_p; \theta) = E(y_j c(\mathfrak{F}_j(y_j), u_1, \ldots, u_p; \theta))$. We make the following assumptions.

**Assumption C.**    (i) *$\dot{c}$ and $c_a$, $a \in \{1, \ldots, p + 1\}$, are continuous.*

(ii) *$E|y_j| < \infty$ for $j \in \{1, \ldots, q\}$.*

(iii) *$E(y_j c_a(\mathfrak{F}_j(y_j), F_1(x_1), \ldots, F_p(x_p); \theta_j^*))^2 < \infty$ and $E(y_j c(\mathfrak{F}_j(y_j), F_1(x_1), \ldots, F_p(x_p); \theta_j^*))^2 < \infty$ for $j \in \{1, \ldots, q\}$ and $a \in \{1, \ldots, p+1\}$.*

(iv) *$E(y_j \frac{\partial c(\mathfrak{F}_j(y_j), F_1(x_1), \ldots, F_p(x_p); \theta_j^*)}{\partial \theta_b})^2 < \infty$ for $j \in \{1, \ldots, q\}$ and $b \in \{1, \ldots, d\}$.*

# 4  Main Results

This section presents the statistical properties of our proposed estimator. The proofs can be found in the Supplementary Materials.

## 4.1  Unbias and Consistency

We first consider the simple case where $p = 1$. Proposition 1 shows that $\Xi_j(x_1) = \vartheta_j(F_1(x_1); \theta_j^*) = E(y_j c(\mathfrak{F}_j(y_j), F_1(x_1); \theta_j^*))$ can be estimated by $\widehat{\Xi}_j(x_1) = \frac{\sum_{i=1}^{n} y_j^{(i)} c(\widehat{\mathfrak{F}}_j(y_j^{(i)}), \widehat{F}_1(x_1); \widehat{\theta}_j)}{n}$. We first provide the following Lemma.

**Lemma 1.** *For $j \in \{1, \ldots, q\}$, suppose that Assumption C holds, if $\widehat{F}_1(x_1) = F_1(x_1) + O_p(n^{-1/2})$, and $\widehat{\theta}_j = \theta_j^* + O_p(n^{-1/2})$, then we have*

$$\widehat{\Xi}_j(x_1) - \frac{\sum_{i=1}^{n} y_j^{(i)} c(\mathfrak{F}_j(y_j^{(i)}), F_1(x_1); \theta_j^*)}{n}$$

$$= 1/n \sum_{i=1}^{n} y_j^{(i)} (\widehat{\mathfrak{F}}_j(y_j^{(i)}) - \mathfrak{F}_j(y_j^{(i)})) c_1(\mathfrak{F}_j(y_j^{(i)}), F_1(x_1); \theta_j^*)$$

$$+ (\widehat{F}_1(x_1) - F_1(x_1)) \vartheta_{j,1}(F_1(x_1); \theta_j^*) + (\widehat{\theta}_j - \theta_j^*)' \dot{\vartheta}_j(F_1(x_1); \theta_j^*) + o_p(n^{-1/2}).$$

Based on Lemma 1, we present the following theorem.

**Theorem 2.** *Given $p = 1$, under Assumptions A, B and the conditions of Lemma 1, for $j \in \{1, \ldots, q\}$, $\widehat{\Xi}_j(x_1)$ is an unbiased and consistent estimator for $\Xi_j(x_1)$.*

In the general case, if $p > 1$, let $F(x) = (F_1(x_1), \ldots, F_p(x_p))$ and $\widehat{F}(x) = (\widehat{F}_1(x_1), \ldots, \widehat{F}_p(x_p))$, $\Xi_j(x) = \frac{\vartheta_j(F(x); \theta_j^*)}{c_x(F(x); \theta_j^*)}$. Similar to the case $p = 1$, we estimate the numerator of $\Xi_j(x)$ by $\widehat{\vartheta}_j(\widehat{F}(x); \widehat{\theta}_j) = 1/n \sum_{i=1}^{n} y_j^{(i)} c(\widehat{\mathfrak{F}}_j(y_j^{(i)}), \widehat{F}(x); \widehat{\theta}_j)$. Under Assumptions A, B and C, suppose that $\widehat{F}_a(x_a) = F_a(x_a) + O_p(n^{-1/2}), \forall a \in \{1, \ldots, p\}$ and $\widehat{\theta}_j = \theta_j^* + O_p(n^{-1/2}), \forall j \in \{1, \ldots, q\}$, according to Lemma 1, we obtain that

$$\widehat{\vartheta}_j(\widehat{F}(x); \widehat{\theta}_j) - \vartheta_j(F(x); \theta_j^*) = 1/n \sum_{i=1}^{n} \Psi_i(x; \theta_j^*) + o_p(n^{-1/2}) \tag{3}$$

where $\Psi_i(x; \theta_j^*) = y_j^{(i)}(\widehat{\mathfrak{F}}_j(y_j^{(i)}) - \mathfrak{F}_j(y_j^{(i)}))c_1(\mathfrak{F}_j(y_j^{(i)}), F(x); \theta_j^*) + \sum_{a=1}^{p}(I(x_a^{(i)} \leq x_a) - F_a(x_a))\vartheta_{j,a}(F(x); \theta_j^*) + \psi_i' \dot{\vartheta}_j(F(x); \theta_j^*)$.

The denominator of $\Xi_j(x)$ can be estimated by $\widehat{c}_x(\widehat{F}(x); \widehat{\theta}_j) = 1/n \sum_{i=1}^{n} c(\widehat{\mathfrak{F}}_j(y_j^{(i)}), \widehat{F}(x); \widehat{\theta}_j)$. Similarly, under conditions of the numerator estimator, we can also obtain that

$$\widehat{c}_x(\widehat{F}(x); \widehat{\theta}_j) - c_x(F(x); \theta_j^*) = 1/n \sum_{i=1}^{n} \mathfrak{W}_i(x; \theta_j^*) + o_p(n^{-1/2}) \tag{4}$$

where $\mathfrak{W}_i(x; \theta_j^*) = \sum_{a=1}^{p}(I(x_a^{(i)} \leq x_a) - F_a(x_a))c_{x,a}(F(x); \theta_j^*) + \psi_i' \dot{c}_x(F(x); \theta_j^*)$.

For $j \in \{1, \ldots, q\}$, the estimator of $\Xi_j(x)$ is given by $\widehat{\Xi}_j(x) = \frac{\widehat{\vartheta}_j(\widehat{F}(x))}{\widehat{c}_x(\widehat{F}(x))} = \frac{\sum_{i=1}^{n} y_j^{(i)} c(\widehat{\mathfrak{F}}_j(y_j^{(i)}), \widehat{F}(x); \widehat{\theta}_j)}{\sum_{i=1}^{n} c(\widehat{\mathfrak{F}}_j(y_j^{(i)}), \widehat{F}(x); \widehat{\theta}_j)}$. Eq.(3) and Eq.(4) lead to the following theorem.

**Theorem 3.** *Under Assumptions A, B and C, suppose that $\widehat{F}_a(x_a) = F_a(x_a) + O_p(n^{-1/2}), \forall a \in \{1, \ldots, p\}$ and $\widehat{\theta}_j = \theta_j^* + O_p(n^{-1/2}), \forall j \in \{1, \ldots, q\}$, we have*

$$\widehat{\Xi}_j(x) - \Xi_j(x) = 1/n \sum_{i=1}^{n} \frac{\Psi_i(x) - \Xi_j(x)\mathfrak{W}_i(x)}{c_x(F(x))} + o_p(n^{-1/2})$$

Based on Theorem 3, we derive the following corollary.

**Corollary 1.** *Given $p > 1$, under conditions of Theorem 3, for $j \in \{1, \ldots, q\}$, $\widehat{\Xi}_j(x)$ is an unbiased and consistent estimator for $\Xi_j(x)$.*

### 4.2 Asymptotic Normality

Let $N(0, 1)$ denote the standard Gaussian distribution. $Var$ represents the variance. We first consider the simple case $p = 1$, based on Lemma 1 and central limit theorem (CLT) [33], we provide the following theorem.

**Theorem 4.** *If $p = 1$, suppose that Assumptions A, B and the conditions of Lemma 1 hold, for $j \in \{1, \ldots, q\}$, we have*

$$\frac{\sqrt{n}(\widehat{\Xi}_j(x_1) - \Xi_j(x_1))}{\sqrt{Var(\Psi_i(x_1))}} \xrightarrow{D} N(0, 1)$$

*where* $\Psi_i(x_1) = y_j^{(i)}(\widehat{\mathfrak{F}}_j(y_j^{(i)}) - \mathfrak{F}_j(y_j^{(i)}))c_1(\mathfrak{F}_j(y_j^{(i)}), F_1(x_1); \theta_j^*) + (I(x_1^{(i)} \leq x_1) - F_1(x_1))\vartheta_{j,1}(F_1(x_1); \theta_j^*) + \psi_i' \dot{\vartheta}_j(F_1(x_1); \theta_j^*)$.

Based on Theorem 3 and CLT, we provide the following theorem in the general case.

**Theorem 5.** *Given $p > 1$, under conditions of Theorem 3, for $j \in \{1, \ldots, q\}$, we have*

$$\frac{\sqrt{n}(\widehat{\Xi}_j(x) - \Xi_j(x))}{\sqrt{Var(\frac{\Psi_i(x) - \Xi_j(x)\mathfrak{W}_i(x)}{c_x(F(x))})}} \xrightarrow{D} N(0, 1)$$

Theorem 5 shows that $\sqrt{n}(\widehat{\Xi}_j(x) - \Xi_j(x))$ follows asymptotically a normal distribution with mean 0 and variance $Var(\frac{\Psi_i(x) - \Xi_j(x)\mathfrak{W}_i(x)}{c_x(F(x))})$.

### 4.3 Bounds on the Mean Squared Error

The mean squared error (MSE) is a basic measure of the accuracy of estimator $\widehat{\Xi}_j(x), \forall j \in \{1, \ldots, q\}$, at an arbitrary fixed point $x = (x_1, \ldots, x_p)' \in \mathbb{R}^{p \times 1}$, which is defined as:

$$MSE_j(x) = E\big(\widehat{\Xi}_j(x) - \Xi_j(x)\big)^2 \tag{5}$$

Eq.(5) can be transformed to the following formula:

$$MSE_j(x) = b_j^2(x) + \sigma_j^2(x) \tag{6}$$

where $b_j(x) = E(\widehat{\Xi}_j(x)) - \Xi_j(x)$ is the bias and $\sigma_j^2(x) = E\left(\widehat{\Xi}_j(x) - E(\widehat{\Xi}_j(x))\right)^2$ is the variance of the estimator $\widehat{\Xi}_j(x)$ at point $x$. §4.1 shows that $\widehat{\Xi}_j(x)$ is an unbiased estimator, so $b_j(x) = 0$. Next, we bound the variance of the estimator. The following theorem is under the case of $p = 1$.

**Theorem 6.** *Given $p = 1$. Suppose that Assumptions A, B and the conditions of Lemma 1 hold. For $j \in \{1, \ldots, q\}$, assume that the density function of $y_j$, $\Upsilon_j(y_j) \leq \Upsilon_{max} < \infty$ and $c(\widehat{\mathfrak{F}}_j(y_j), \widehat{F}_1(x_1)) \leq c_{max} < \infty$, then we have*

$$MSE_j(x_1) \leq \frac{7\Upsilon_{max}c_{max}^2}{6n}$$

Similarly, we obtain the following theorem in the general case.

**Theorem 7.** *Given $p > 1$. Suppose that the conditions of Theorem 3 hold. For $j \in \{1, \ldots, q\}$, assume that the density function of $y_j$, $\Upsilon_j(y_j) \leq \Upsilon_{max} < \infty$, then we have*

$$MSE_j(x) \leq \frac{7\Upsilon_{max}}{6n}$$

**Remark.** Theorem 6 and Theorem 7 show that the MSE of $\widehat{\Xi}_j(x_1)$ and $\widehat{\Xi}_j(x), \forall j \in \{1, \ldots, q\}$, go to 0 as $n$ goes to infinity.

## 5 Experiment

### 5.1 Data Sets and Baselines

This paper considers two popular families of copulas:

- Multivariate normal copula: A Gaussian copula with a given correlation matrix $Z \in [-1, 1]^{d \times d}$ is defined as $C(u_1, \ldots, u_d; Z) = \Phi_Z(\Phi^{-1}(u_1), \ldots, \Phi^{-1}(u_d))$, where $\Phi^{-1}$ is the inverse CDF of a standard normal distribution and $\Phi_Z$ is the joint CDF of a multivariate normal distribution with zero mean vector and correlation matrix equal to $Z$. The density function is written as $c(u_1, \ldots, u_d; Z) = det(Z)^{-1/2}exp(-1/2\varsigma'(Z^{-1} - I_d)\varsigma)$, where $det(Z)$ represents the determinant of $Z$, $\varsigma = (\Phi^{-1}(u_1), \ldots, \Phi^{-1}(u_d))'$ and $I_d$ is the identity matrix.

- Multivariate student's t copula: A student's t copula with a given correlation matrix $Z \in [-1, 1]^{d \times d}$ and degree of freedom $\nu$ is defined as $C(u_1, \ldots, u_d; Z, \nu) = T_{Z,\nu}(T_\nu^{-1}(u_1), \ldots, T_\nu^{-1}(u_d))$, where $T_\nu^{-1}$ is the inverse CDF of a student's t-distribution with degree of freedom $\nu$ and $T_{Z,\nu}$ is the joint CDF of a multivariate student's t-distribution with a correlation matrix $Z$ and degree of freedom $\nu$. The density function is $c(u_1, \ldots, u_d; Z, \nu) = det(Z)^{-1/2}\frac{\Gamma((\nu+d)/2)(\Gamma(\nu/2))^{d-1}}{(\Gamma((\nu+1)/2))^d}\frac{(1+1/\nu\varsigma'Z^{-1}\varsigma)^{-(\nu+d)/2}}{\prod_{i=1}^d(1+\varsigma_i^2/\nu)^{-(\nu+1)/2}}$, where $\Gamma$ is the gamma function, $\varsigma = (\varsigma_1, \ldots, \varsigma_d)'$ with $\varsigma_i = T_\nu^{-1}(u_i), \forall i = 1, \ldots, d$.

Table 1: The results of Example-F1 on the various data sets (mean ± standard deviation). The best ones are in bold.

| DATA SET | BR | CC | CCA | CPLST | CML+GAU | CML+ST |
|---|---|---|---|---|---|---|
| EMOTIONS | $0.5051 \pm 0.0288$ | $0.5063 \pm 0.0349$ | $0.5141 \pm 0.0283$ | $0.5178 \pm 0.0686$ | $\mathbf{0.5215} \pm 0.0642$ | $0.5146 \pm 0.0389$ |
| SCENE | $0.5203 \pm 0.0509$ | $0.5358 \pm 0.0229$ | $0.5212 \pm 0.0559$ | $0.5311 \pm 0.0533$ | $\mathbf{0.5378} \pm 0.0250$ | $0.5367 \pm 0.0571$ |
| MEDICAL | $0.1633 \pm 0.0738$ | $0.1690 \pm 0.0145$ | $0.1713 \pm 0.0866$ | $0.1716 \pm 0.0372$ | $\mathbf{0.1767} \pm 0.0160$ | $0.1728 \pm 0.0551$ |
| YEAST | $0.5578 \pm 0.0153$ | $0.5620 \pm 0.0329$ | $0.5609 \pm 0.0124$ | $0.5649 \pm 0.0323$ | $0.5647 \pm 0.0112$ | $\mathbf{0.5893} \pm 0.0429$ |
| ENRON | $0.1707 \pm 0.0536$ | $0.1799 \pm 0.0107$ | $0.1756 \pm 0.0413$ | $0.1822 \pm 0.0225$ | $\mathbf{0.1888} \pm 0.0313$ | $0.1831 \pm 0.0322$ |

Table 2: The results of Macro-F1 on the various data sets (mean ± standard deviation). The best ones are in bold.

| DATA SET | BR | CC | CCA | CPLST | CML+GAU | CML+ST |
|---|---|---|---|---|---|---|
| EMOTIONS | $0.5475 \pm 0.0172$ | $0.5510 \pm 0.0304$ | $0.5492 \pm 0.0259$ | $0.5528 \pm 0.0393$ | $\mathbf{0.5701} \pm 0.0147$ | $0.5534 \pm 0.0355$ |
| SCENE | $0.5801 \pm 0.0540$ | $0.5814 \pm 0.0163$ | $0.5910 \pm 0.0335$ | $0.5926 \pm 0.0312$ | $\mathbf{0.6084} \pm 0.0217$ | $0.5953 \pm 0.0411$ |
| MEDICAL | $0.0669 \pm 0.0417$ | $0.0769 \pm 0.0153$ | $0.0677 \pm 0.0276$ | $0.0733 \pm 0.0463$ | $0.0751 \pm 0.0018$ | $\mathbf{0.0808} \pm 0.0199$ |
| YEAST | $0.3342 \pm 0.0115$ | $0.3393 \pm 0.0267$ | $0.3373 \pm 0.0106$ | $0.3453 \pm 0.0381$ | $0.3523 \pm 0.0143$ | $\mathbf{0.3590} \pm 0.0145$ |
| ENRON | $0.0165 \pm 0.0054$ | $0.0176 \pm 0.0018$ | $0.0216 \pm 0.0033$ | $0.0194 \pm 0.0016$ | $0.0218 \pm 0.0016$ | $\mathbf{0.0220} \pm 0.0030$ |

We abbreviate our proposed copula multi-label learning with multivariate normal copula and multivariate student's t copula to CML+GAU and CML+ST, respectively. This section evaluates the performance of the proposed method on five real-world benchmark data sets with various domains: EMOTIONS (music), SCENE (image), MEDICAL (text), YEAST (biology) and ENRON (text). The statistics of these data sets are presented in the website[1]. We compare our CML+GAU and CML+ST with several multi-label learning approaches, which aim to capture the interdependencies between labels: BR, CC, CCA and CPLST. We use the code provided by the respective authors with default parameters. The bandwidth is set to $h = 0.1$ in the experiment.

To fairly measure the performance of our methods and the baseline methods, we consider Hamming Loss, Example-F1, Micro-F1 and Macro-F1 as the evaluation measurements [10, 34, 35]. The smaller the value of Hamming Loss, the better the performance, while the larger the value of the other three measurements, the better the performance. We perform 3-fold cross-validation on each data set and report the mean and standard error of each evaluation measurement.

## 5.2 Results

Tables 1 and 2 show the Example-F1 and Macro-F1 results for our methods and baseline approaches in respect of the different data sets. The Hamming Loss and Micro-F1 results are reported in the Supplementary Materials. From Tables 1 and 2, we can see that: 1) BR generally underperforms. BR does not consider the distributions and relationships between labels, so it achieves much lower accuracy. 2) CPLST outperforms CC and CCA, because CPLST captures both the label and the feature dependency. 3) Our proposed methods are most successful on all data sets. The empirical results illustrate the superiority of our proposed model and corroborate our theoretical studies.

## 6 Conclusion

The great success of copulas in a wide range of applications inspires us to develop a novel copula multi-label learning paradigm for modeling label and feature dependencies, and reveal new statistical insights in multi-label learning. Particularly, after leveraging the kernel trick to construct continuous distribution in the output space, we use a semiparametric approach to estimate our proposed model. Theoretically, we show that our estimator is an unbiased and consistent estimator and the distribution of our proposed estimator converges to a normal distribution. Moreover, we provide the bound for the MSE of estimator. The experimental results demonstrate the effectiveness of our proposed approach.

### Acknowledgments

This work is supported by the National Natural Science Foundation of China under Grants 61976161 and the Science and Technology Major Project of Hubei Province (Next-Generation AI Technologies).

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
