[Supplementary Material]

# Copula Multi-label Learning (Supplementary)

**Weiwei Liu**
School of Computer Science, Wuhan University
Wuhan, China 430072
liuweiwei863@gmail.com

## Abstract

In this supplementary file, we first present the proofs of some important propositions, lemmas and theorems in the main paper. After that, we present more experiment results.

## 1 Assumptions

**Assumption A.** *For $j \in \{1, \ldots, p\}$,*

$$\widehat{F}_j(x_j) = \frac{\sum_{i=1}^n I(x_j^{(i)} \leq x_j)}{n} + o_p(n^{-1/2})$$

**Assumption B.** *For $j \in \{1, \ldots, q\}$,*

$$\widehat{\theta}_j - \theta_j^* = \frac{\sum_{i=1}^n \psi_i}{n} + o_p(n^{-1/2})$$

*where $\psi_i = \psi(\mathfrak{F}_j(y_j^{(i)}), F_1(x_1^{(i)}), \ldots, F_p(x_p^{(i)}); \theta_j^*)$ is a $d$-dimensional random vector such that $E(\psi) = (0, \ldots, 0)'$ and $E\|\psi\|_2^2 < \infty$.*

**Assumption C.**     *(i) $\dot{c}$ and $c_a$, $a \in \{1, \ldots, p+1\}$, are continuous.*

   *(ii) $E|y_j| < \infty$ for $j \in \{1, \ldots, q\}$.*

   *(iii) $E(y_j c_a(\mathfrak{F}_j(y_j), F_1(x_1), \ldots, F_p(x_p); \theta_j^*))^2 \qquad < \qquad \infty \qquad$ and*
   *$E(y_j c(\mathfrak{F}_j(y_j), F_1(x_1), \ldots, F_p(x_p); \theta_j^*))^2 < \infty$ for $j \in \{1, \ldots, q\}$ and $a \in \{1, \ldots, p+1\}$.*

   *(iv) $E(y_j \frac{\partial c(\mathfrak{F}_j(y_j), F_1(x_1), \ldots, F_p(x_p); \theta_j^*)}{\partial \theta_b})^2 < \infty$ for $j \in \{1, \ldots, q\}$ and $b \in \{1, \ldots, d\}$.*

## 2 Proof of Proposition 1

**Proposition 1.** *If $p = 1$ or $x_1, \ldots, x_p$ are mutually independent, Eq.(2) of the main paper reduces to $\Xi_j(x) = \vartheta_j(F_1(x_1), \ldots, F_p(x_p)), \forall j \in \{1, \ldots, q\}$.*

*Proof.* For $j \in \{1, \ldots, q\}$, if $p = 1$, $C(1, u_1) = P(\mathfrak{F}_j(y_j) <= 1, F_1(x_1) \leq u_1) = P(F_1(x_1) \leq u_1)$. The CDF of a continuous variable has the uniform distribution, so $C(1, u_1) = u_1$ and $c_x(u_1) = 1$. If $x_1, \ldots, x_p$ are mutually independent, $C(1, u_1, \ldots, u_p) = P(\mathfrak{F}_j(y_j) <= 1, F_1(x_1) \leq u_1, \ldots, F_p(x_p) \leq u_p) = P(F_1(x_1) \leq u_1) \times \ldots \times P(F_p(x_p) \leq u_p) = u_1 \times \ldots \times u_p$, so $c_x(F_1(x_1), \ldots, F_p(x_p)) = 1$. □

## 3  Proof of Lemma 1

**Lemma 1.** *For $j \in \{1, \ldots, q\}$, suppose that Assumption C holds, if $\widehat{F}_1(x_1) = F_1(x_1) + O_p(n^{-1/2})$, and $\widehat{\theta}_j = \theta_j^* + O_p(n^{-1/2})$, then we have*

$$
\widehat{\Xi}_j(x_1) - \frac{\sum_{i=1}^n y_j^{(i)} c(\mathfrak{F}_j(y_j^{(i)}), F_1(x_1); \theta_j^*)}{n}
$$
$$
= 1/n \sum_{i=1}^n y_j^{(i)} (\widehat{\mathfrak{F}}_j(y_j^{(i)}) - \mathfrak{F}_j(y_j^{(i)})) c_1(\mathfrak{F}_j(y_j^{(i)}), F_1(x_1); \theta_j^*)
$$
$$
+ (\widehat{F}_1(x_1) - F_1(x_1)) \vartheta_{j,1}(F_1(x_1); \theta_j^*) + (\widehat{\theta}_j - \theta_j^*)' \dot{\vartheta}_j(F_1(x_1); \theta_j^*) + o_p(n^{-1/2}).
$$

*Proof.* Given $j \in \{1, \ldots, q\}$, using Taylor expansion, we have

$$
\widehat{\Xi}_j(x_1) = \frac{\sum_{i=1}^n y_j^{(i)} c(\mathfrak{F}_j(y_j^{(i)}), F_1(x_1); \theta_j^*)}{n} + \Lambda_1 + \Lambda_2 + + \Lambda_3 \tag{1}
$$

where

$$
\Lambda_1 = 1/n \sum_{i=1}^n y_j^{(i)} (\widehat{\mathfrak{F}}_j(y_j^{(i)}) - \mathfrak{F}_j(y_j^{(i)})) c_1(\widetilde{u}_{i,j}, \widetilde{u}_1; \widetilde{\theta}_j)
$$

$$
\Lambda_2 = 1/n \sum_{i=1}^n y_j^{(i)} (\widehat{F}_1(x_1) - F_1(x_1)) c_2(\widetilde{u}_{i,j}, \widetilde{u}_1; \widetilde{\theta}_j)
$$

$$
\Lambda_3 = 1/n \sum_{i=1}^n y_j^{(i)} (\widehat{\theta}_j - \theta_j^*)' \dot{c}(\widetilde{u}_{i,j}, \widetilde{u}_1; \widetilde{\theta}_j)
$$

with $\widetilde{u}_{i,j} = \mathfrak{F}_j(y_j^{(i)}) + t(\widehat{\mathfrak{F}}_j(y_j^{(i)}) - \mathfrak{F}_j(y_j^{(i)}))$, $\widetilde{u}_1 = F_1(x_1) + t(\widehat{F}_1(x_1) - F_1(x_1))$ and $\widetilde{\theta}_j = \theta_j^* + t(\widehat{\theta}_j - \theta_j^*)$ for some $t \in [0, 1]$. $\Lambda_1$ can be represented as

$$
\Lambda_1 = 1/n \sum_{i=1}^n y_j^{(i)} (\widehat{\mathfrak{F}}_j(y_j^{(i)}) - \mathfrak{F}_j(y_j^{(i)})) \times c_1(\mathfrak{F}_j(y_j^{(i)}), F_1(x_1); \theta_j^*) + R_1
$$

where

$$
R_1 = 1/n \sum_{i=1}^n y_j^{(i)} (\widehat{\mathfrak{F}}_j(y_j^{(i)}) - \mathfrak{F}_j(y_j^{(i)})) \big( c_1(\widetilde{u}_{i,j}, \widetilde{u}_1; \widetilde{\theta}_j) - c_1(\mathfrak{F}_j(y_j^{(i)}), F_1(x_1); \theta_j^*) \big)
$$

Assumption C.(ii) shows that $E|y_j| < \infty$. We get that $|y_j|$ has a constant upper bound, so $1/n \sum_{i=1}^n |y_j^{(i)}| = O_p(1)$. From De Moivre's theorem [1], we obtain that $\widehat{p}_j(0) - p_j(0) = O_p(n^{-1/2})$ and $\widehat{p}_j(1) - p_j(1) = O_p(n^{-1/2})$, so $\sup_{y_j} |\widehat{\mathfrak{F}}_j(y_j) - \mathfrak{F}_j(y_j)| = O_p(n^{-1/2})$. By Assumption C.(i) and the continuous mapping theorem [2], we have $\sup_i |c_1(\widetilde{u}_{i,j}, \widetilde{u}_1; \widetilde{\theta}_j) - c_1(\mathfrak{F}_j(y_j^{(i)}), F_1(x_1); \theta_j^*)| = o_p(1)$. Then we have

$$
|R_1| \le 1/n \sum_{i=1}^n |y_j^{(i)}| \sup_{y_j} |\widehat{\mathfrak{F}}_j(y_j^{(i)}) - \mathfrak{F}_j(y_j^{(i)})| \sup_i |c_1(\widetilde{u}_{i,j}, \widetilde{u}_1; \widetilde{\theta}_j) - c_1(\mathfrak{F}_j(y_j^{(i)}), F_1(x_1); \theta_j^*)|
$$
$$
\le O_p(1) O_p(n^{-1/2}) o_p(1) = o_p(n^{-1/2})
$$

Thus, we obtain

$$
\Lambda_1 = 1/n \sum_{i=1}^n y_j^{(i)} (\widehat{\mathfrak{F}}_j(y_j^{(i)}) - \mathfrak{F}_j(y_j^{(i)})) c_1(\mathfrak{F}_j(y_j^{(i)}), F_1(x_1); \theta_j^*) + o_p(n^{-1/2}) \tag{2}
$$

Similarly, if $\widehat{F}_1(x_1) = F_1(x_1) + O_p(n^{-1/2})$, we have

$$
\Lambda_2 = 1/n \sum_{i=1}^n y_j^{(i)} (\widehat{F}_1(x_1) - F_1(x_1)) c_2(\mathfrak{F}_j(y_j^{(i)}), F_1(x_1); \theta_j^*) + o_p(n^{-1/2})
$$

Using Assumption C.(iii) and weak law of large numbers (WLLN) [1], we know that $\frac{\sum_{i=1}^{n} y_j^{(i)} c_2(\mathfrak{F}_j(y_j^{(i)}), F_1(x_1); \theta_j^*)}{n} \xrightarrow{P} E(y_j c_2(\mathfrak{F}_j(y_j), F_1(x_1); \theta_j^*)) = \vartheta_{j,1}(F_1(x_1); \theta_j^*)$, so we obtain

$$\Lambda_2 = (\widehat{F}_1(x_1) - F_1(x_1))\vartheta_{j,1}(F_1(x_1); \theta_j^*) + o_p(n^{-1/2}) \tag{3}$$

Using $\widehat{\theta}_j = \theta_j^* + O_p(n^{-1/2})$, Assumption C.(iv) and WLLN again, we have

$$\begin{aligned}
\Lambda_3 &= 1/n \sum_{i=1}^{n} y_j^{(i)} (\widehat{\theta}_j - \theta_j^*)' \dot{c}(\mathfrak{F}_j(y_j^{(i)}), F_1(x_1); \theta_j^*) + o_p(n^{-1/2}) \\
&= (\widehat{\theta}_j - \theta_j^*)' \dot{\vartheta}_j(F_1(x_1); \theta_j^*) + o_p(n^{-1/2})
\end{aligned} \tag{4}$$

Combining Eq.(1), Eq.(2), Eq.(3) and Eq.(4) implies the result. □

We first consider the simple case where $p = 1$. Proposition 1 shows that $\Xi_j(x_1) = \vartheta_j(F_1(x_1); \theta_j^*) = E(y_j c(\mathfrak{F}_j(y_j), F_1(x_1); \theta_j^*))$ can be estimated by $\widehat{\Xi}_j(x_1) = \frac{\sum_{i=1}^{n} y_j^{(i)} c(\widehat{\mathfrak{F}}_j(y_j^{(i)}), \widehat{F}_1(x_1); \widehat{\theta}_j)}{n}$. We first provide the following Lemma.

## 4 Proof of Theorem 2

**Theorem 2.** *Given $p = 1$, under Assumptions A, B and the conditions of Lemma 1, for $j \in \{1, \ldots, q\}$, $\widehat{\Xi}_j(x_1)$ is an unbiased and consistent estimator for $\Xi_j(x_1)$.*

*Proof.* For $j \in \{1, \ldots, q\}$, from Lemma 1, we know that

$$\begin{aligned}
E\left(\widehat{\Xi}_j(x_1)\right) =& E\left(\frac{\sum_{i=1}^{n} y_j^{(i)} c(\mathfrak{F}_j(y_j^{(i)}), F_1(x_1); \theta_j^*)}{n}\right) \\
&+ E\left(1/n \sum_{i=1}^{n} y_j^{(i)} (\widehat{\mathfrak{F}}_j(y_j^{(i)}) - \mathfrak{F}_j(y_j^{(i)})) c_1(\mathfrak{F}_j(y_j^{(i)}), F_1(x_1); \theta_j^*)\right) \\
&+ E\left((\widehat{F}_1(x_1) - F_1(x_1))\vartheta_{j,1}(F_1(x_1); \theta_j^*)\right) + E\left((\widehat{\theta}_j - \theta_j^*)' \dot{\vartheta}_j(F_1(x_1); \theta_j^*)\right)
\end{aligned} \tag{5}$$

Now, we deal with each term in the right side of Eq.(5).

$$E\left(\frac{\sum_{i=1}^{n} y_j^{(i)} c(\mathfrak{F}_j(y_j^{(i)}), F_1(x_1); \theta_j^*)}{n}\right) = E(y_j c(\mathfrak{F}_j(y_j), F_1(x_1); \theta_j^*)) = \Xi_j(x_1)$$

If $-0.5 \le y_j \le 0.5$, using the law of total expectation, we obtain that

$$\begin{aligned}
&E\left(y_j (\widehat{\mathfrak{F}}_j(y_j) - \mathfrak{F}_j(y_j)) c_1(\mathfrak{F}_j(y_j), F_1(x_1); \theta_j^*)\right) \\
=& E\left[E\left(y_j (\widehat{\mathfrak{F}}_j(y_j) - \mathfrak{F}_j(y_j)) c_1(\mathfrak{F}_j(y_j), F_1(x_1); \theta_j^*)|y_j\right)\right] \\
=& E\left[E\left(y_j (\frac{\sum_{a=1}^{n} I(y_j^a = 0)}{n}(y_j + 0.5) - \mathfrak{F}_j(y_j)) c_1(\mathfrak{F}_j(y_j), F_1(x_1); \theta_j^*)|y_j\right)\right] \\
=& E\left[y_j \left(p_j(0)(y_j + 0.5) - \mathfrak{F}_j(y_j)\right) c_1(\mathfrak{F}_j(y_j), F_1(x_1); \theta_j^*)\right] = 0
\end{aligned}$$

If $0.5 < y_j \le 1.5$, similarly, we obtain that $E\left(y_j (\widehat{\mathfrak{F}}_j(y_j) - \mathfrak{F}_j(y_j)) c_1(\mathfrak{F}_j(y_j), F_1(x_1); \theta_j^*)\right) = 0$.

Using Assumptions A and B, we obtain that $E\left((\widehat{F}_1(x_1) - F_1(x_1))\vartheta_{j,1}(F_1(x_1); \theta_j^*)\right) = 0$ and $E\left((\widehat{\theta}_j - \theta_j^*)' \dot{\vartheta}_j(F_1(x_1); \theta_j^*)\right) = 0$. Thus, $E\left(\widehat{\Xi}_j(x_1)\right) = \Xi_j(x_1)$, and from Assumption C.(iii) and WLLN, we know that $\widehat{\Xi}_j(x_1)$ is an unbiased and consistent estimator for $\Xi_j(x_1)$. □

Table 1: The results of Hamming Loss on the various data sets (mean $\pm$ standard deviation). The best ones are in bold.

| DATA SET | BR | CC | CCA | CPLST | CML+GAU | CML+ST |
|---|---|---|---|---|---|---|
| EMOTIONS | $0.2628 \pm 0.0155$ | $0.2590 \pm 0.0214$ | $0.2607 \pm 0.0408$ | $0.2575 \pm 0.0448$ | $\mathbf{0.2435} \pm 0.0208$ | $0.2500 \pm 0.0213$ |
| SCENE | $0.1483 \pm 0.0109$ | $0.1367 \pm 0.0078$ | $0.1342 \pm 0.0538$ | $0.1331 \pm 0.0321$ | $\mathbf{0.1315} \pm 0.0073$ | $0.1333 \pm 0.0101$ |
| MEDICAL | $0.1293 \pm 0.0076$ | $0.1292 \pm 0.0166$ | $0.1259 \pm 0.0578$ | $0.1273 \pm 0.0288$ | $0.1250 \pm 0.0174$ | $\mathbf{0.1170} \pm 0.0070$ |
| YEAST | $0.2586 \pm 0.0186$ | $0.2521 \pm 0.0142$ | $0.2558 \pm 0.0213$ | $0.2472 \pm 0.0093$ | $\mathbf{0.2403} \pm 0.0036$ | $0.2467 \pm 0.0226$ |
| ENRON | $0.0659 \pm 0.0031$ | $0.0655 \pm 0.0019$ | $0.0642 \pm 0.0220$ | $0.0637 \pm 0.0051$ | $0.0626 \pm 0.0062$ | $\mathbf{0.0613} \pm 0.0042$ |

Table 2: The results of Micro-F1 on the various data sets (mean $\pm$ standard deviation). The best ones are in bold.

| DATA SET | BR | CC | CCA | CPLST | CML+GAU | CML+ST |
|---|---|---|---|---|---|---|
| EMOTIONS | $0.5605 \pm 0.0222$ | $0.5651 \pm 0.0359$ | $0.5612 \pm 0.0228$ | $0.5716 \pm 0.0588$ | $\mathbf{0.5794} \pm 0.0362$ | $0.5744 \pm 0.0400$ |
| SCENE | $0.5966 \pm 0.0518$ | $0.6004 \pm 0.0214$ | $0.6021 \pm 0.0282$ | $0.6026 \pm 0.0691$ | $\mathbf{0.6118} \pm 0.0267$ | $0.6022 \pm 0.0150$ |
| MEDICAL | $0.2475 \pm 0.0940$ | $0.2491 \pm 0.0127$ | $0.2556 \pm 0.0628$ | $0.2551 \pm 0.0415$ | $0.2568 \pm 0.0215$ | $\mathbf{0.2582} \pm 0.0595$ |
| YEAST | $0.5761 \pm 0.0206$ | $0.5853 \pm 0.0291$ | $0.5778 \pm 0.0212$ | $0.5885 \pm 0.0372$ | $\mathbf{0.5962} \pm 0.0042$ | $0.5862 \pm 0.0308$ |
| ENRON | $0.1863 \pm 0.0494$ | $0.1909 \pm 0.0190$ | $0.1877 \pm 0.0690$ | $0.1922 \pm 0.0358$ | $0.1943 \pm 0.0124$ | $\mathbf{0.1961} \pm 0.0289$ |

## 5 Proof of Theorem 6

**Theorem 6.** *Given $p = 1$. Suppose that Assumptions A, B and the conditions of Lemma 1 hold. For $j \in \{1, \ldots, q\}$, assume that the density function of $y_j$, $\Upsilon_j(y_j) \leq \Upsilon_{max} < \infty$ and $c(\widehat{\mathfrak{F}}_j(y_j), \widehat{F}_1(x_1)) \leq c_{max} < \infty$, then we have*

$$MSE_j(x_1) \leq \frac{7\Upsilon_{max}c_{max}^2}{6n}$$

*Proof.* For $j \in \{1, \ldots, q\}$, set

$$\kappa_{i,j} = y_j^{(i)} c(\widehat{\mathfrak{F}}_j(y_j^{(i)}), \widehat{F}_1(x_1)) - E\big(y_j^{(i)} c(\widehat{\mathfrak{F}}_j(y_j^{(i)}), \widehat{F}_1(x_1))\big)$$

Then $\kappa_{i,j}$, $i = 1, \ldots, n$, are $n$ i.i.d. random variables with zero mean and variance

$$E(\kappa_{i,j}^2) \leq E\Big(\big(y_j^{(i)} c(\widehat{\mathfrak{F}}_j(y_j^{(i)}), \widehat{F}_1(x_1))\big)^2\Big) \leq c_{max}^2 E(y_j^{(i)})^2$$

$$\leq c_{max}^2 \int (y_j)^2 \Upsilon_j(y_j) dy_j \leq \frac{7c_{max}^2 \Upsilon_{max}}{6}$$

Then, we obtain

$$MSE_j(x_1) = \sigma_j^2(x_1) = E\Big(\big(\frac{\sum_{i=1}^n \kappa_{i,j}}{n}\big)^2\Big) = \frac{E(\kappa_{i,j}^2)}{n} \leq \frac{7c_{max}^2 \Upsilon_{max}}{6n}$$

$\square$

## 6 More Results

This section shows more experiment results. Tables 1 and 2 list the Hamming Loss and Micro-F1 results for our methods and baseline approaches in respect of the different data sets. From Tables 1 and 2, we can see that our proposed methods achieve the best performance.