[Reviews · NeurIPS 2019]

Reviewer 1



This paper proposes copula multi-label learning to explain the statistical properties of multi-label dependency. Copula has been widely used for multivariate data in many areas. To use copula in multi-label learning, the paper first leverages the kernel trick to estimate the multi-label distribution. The proposed model is estimated semiparametrically. Theoretical analysis is provided, and the error bound of the estimator is also given. The experiment validates the proposed method is better than existing methods. Originality: The paper is original. Quality: The paper appears to be of high quality and contains a substantial theoretical advance. Clarity: The exposition is clear. Significance: This work is likely of significance to the community. I suggest acceptance of the paper.

Reviewer 2



[Reply to authors feedback] I thank the authors for their answers. After that and reading the other reviews, I maintain that while the approach is original and potentially interesting, I miss a deeper analysis of the joint model. In particular: * There is no clear theoretical justification to develop a complete joint complex model if it is then to focus on the marginal distributions (whose estimation does not require estimation of the dependencies), and in my viewpoint the biggest benefit of having a full joint distribution is the ability to optimize arbitrary loss function, a topic never touched upon in the paper. * I miss a comparison with other approaches, not based on copulas, that also try to completely model the dependencies between labels, such as probabilistic chaining, be it from a theoretical or empirical perspective. Originality: to my knowledge, using copulas to solve the global multi-label problem has not been done before, hence the paper can be considered as original. Quality: the submission appears to be correct, yet I honestly did not dwelve into the formulas, as the presentation is quite heavy in notations, starting with the presentation that is very technical, and much more involuted than classical presentation of copulas. Being familiar with copulas, I could follow most of the explanations, but someone unfamaliar with them would probably have a quite hard time to follow the paper. It is also surprising that on the one hand authors try to identify the whole joint distribution, if it is at the end to make their predictions by only focusing on the marginals on the labels (L140-L144), which in theory (but in practice is) is not different from BR. There are no discussion related to optiizing particular loss functions (in particular the one used in the experiments), nor of the connection with the probabilistic chaining, that in principle can minimize any given loss. It is somehow disppointing to use a fancy model to then make predictions using only the marginals on the labels. What about commparing with methods trying to optimize such loss functions? The experiments are also a bit disappointing, in the sense tha tthe consistent improvement observed is a bit low, and almost always within the standard deviation interval given in the results. A lot of efforts is spent on showing good properties of the estimator, however it is not clear to me whether a simple BR estimator also does not enjoy such similar properties, at least of unbiasedness and convergence of the marginal estimate? Clarity: the paper is well-written, but the mathematics are written in a quite complex way. I think this could be simplified

Reviewer 3



Due to wide applications, multi-label learning has become one of the most important research area in machine learning. However, the statistical properties of existing multi-label dependency modelings are not well understood. To provide the statistical understandings for multi-label learning, this paper proposes a new copula multi-label learning paradigm for modeling label and feature dependencies. The continuous distribution in the output space is first constructed by leveraging the kernel trick in this paper, and then the proposed model is estimated semi-parametrically. Moreover, this paper shows that the proposed estimator is an unbiased and consistent estimator and follows asymptotically a normal distribution, and they further provide the bound for the mean squared error of estimator. The experiment validates the performance of the proposed method.

[Author Response · NeurIPS 2019]

Table 1: The results of Example-F1 on the EMOTIONS, SCENE and MEDICAL data sets.

| DATA SET | LIMO | CML+GAU |
|----------|------|---------|
| EMOTIONS | 0.5216 | 0.5215 |
| SCENE | 0.5376 | 0.5378 |
| MEDICAL | 0.1769 | 0.1767 |

**Response to Reviewer #1**

**Q1: What is the performance of the proposed method on the image datasets? A:** Thanks for your comments. We have conducted the experiment on the image dataset (scene). Moreover, we also test our methods on other data sets with different domains, such as text (medical, enron), biology (yeast) and music (emotions). The experiment results have shown the improved performance of our proposed methods on various domains.

**Response to Reviewer #2**

**Q1: . . . focusing on the marginals. . . which in theory (but in practice is) is not different from BR . . . ? A:** Thanks for your comments. Theorem 1 indicates that copula allows a complete separation of dependence modeling from the marginal distributions and by specifying a copula one can summarize all the dependencies between margins.

Therefore, based on Theorem 1, we can use a $(p + q)$-copula function to summarize all the dependencies between labels and features. Based on $(p + q)$-copula function, we are able to derive marginal functions. Please note that the derived marginal functions have already inherited the dependence information from $(p + q)$-copula function. We have presented this point in lines 125-129.

**Q2: . . . loss functions. . . used in the experiments. . . comparison with F1-measure optimizers. . . A:** Thanks for your comments. This paper optimizes the Hamming loss, and in the Supplementary Materials, Table 1 shows the results of Hamming loss on the various data sets.

LIMO (A Unified View of Multi-Label Performance Measures, ICML, 2017) is a state-of-the-art F1-measure optimizer. According to the reviewer's comments, we compare our proposed method with LIMO in terms of Example-F1 on the EMOTIONS, SCENE and MEDICAL data sets. The results are shown in Tabel 1. From Tabel 1, we can see that our proposed method is comparable to the F1-measure optimizer. Following the reviewer's comments, we will consider optimizing various loss functions in the future work.

**Q3: whether a simple BR estimator also does not enjoy such similar properties. A:** Thanks for your comments. One of the most important insights of Sklar's Theorem (Theorem 1) is that the univariate margins and the multivariate dependence structure can be separated, and the dependence structure can be represented by a copula. Therefore, by specifying a copula one can summarize all the dependencies between margins. Inspired by Sklar's Theorem, we develop a framework of copula multi-label learning to model label and feature dependencies. The theoretical analysis in this paper makes no assumptions on the specific copula functions. We can derive the same statistical properties for our proposed estimator with any copula functions. If there is a copula contain the independent information between the labels, then our theoretical results also hold in this special case (BR).

**Response to Reviewer #3**

**Q1: This paper uses normal copula and student's copula, is it possible to use other copula functions? A:** Thanks for your comments. The theoretical analysis in this paper makes no assumptions on the specific copula functions. We can derive the same statistical properties for our proposed estimator with any copula functions. In the experiment, we use multivariate normal copula and multivariate student's t copula as two examples to show the performance of our proposed method.

**Q2: The paper may need to add and discuss the three references. A:** Thanks for your comments. These papers focus on the applications, such as image classification, text classification and health evaluation. We will cite these references in the revisions.



[Meta-Review · NeurIPS 2019]

This paper presents a copula-based approach to multi-label multi-class classification. Under some conditions it is shown that the estimated conditional expectation is unbiased and bounds on the Mean Square Error are derived. The paper is well written and I found the analyzes and derivations correct. The use of copula for the study of the multi-label setting is new and the study can help to orient more theoretically driven works in this direction.